# Electric Vehicles and Psychology

Fabio Viola 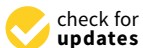

Dipartimento d'Ingegneria, University of Palermo, 90128 Palermo, Italy; fabio.viola@unipa.it

**Abstract:** The popularity of electric vehicles is evidenced by the broad range of manufacturers presenting new models of plug-in hybrid and battery vehicles. However, the success of the revolution or, rather, the rebirth of electric vehicles, is hanging by a thread, as it lacks the involvement of a large number of users, and many psychological mechanisms hinder it. What are users' true feelings about this new world of vehicles? Are people ready for the fifth level of automation, i.e., fully automatic driving and the absence of the driving position? The purpose of this paper is to present and discuss the psychological aspects that influence the adoption of electric vehicles. Topics such as the chicken and egg paradox (electric vehicles and charging stations) and performance anxiety (regarding, e.g., range) are addressed. This review is characterized by contradictions and irony.

**Keywords:** electric vehicles; range anxiety; chicken and egg paradox; battery; public perception; key motivators and barriers

## 1. Introduction

Psychology is an integral part of our daily life. The perception of reality is the result of conditioning, which may be positive. The car user is inundated with information that valorizes aspects that often have little to do with driving itself.

On the other hand, the ideology of the present reigns, which tries to ensure the preservation of the status quo. An ideology that can halt progress is thinking of the present as the child of the past, i.e., in which the teachings of the past are obsolete and desire for the future is neglected in favor of a more comfortable present. The scientific aspect is often self-centered, if not hegemonic. The public perception of electric cars is that they are low-range vehicles due to their limited battery capacity, and not that electricity is more widely available than gasoline.

The autonomy of a vehicle with a battery is proportional to the capacity of the battery, which is proportional to the cost of the vehicle (i.e., around 40–48% of the cost of a vehicle is dependent upon the quality of its battery). This is a scientific way of thinking that does not foresee why or how to use battery power, but only exalts the solution: To buy a vehicle with greater autonomy that is more expensive. The present is hegemonic, and the user often considers battery swapping as a definitive solution to the limitations of battery use. This idea represents a transposition of the present into a future, and not a future means of transportation.

Can a different approach change the rules of the game, or will we be prisoners of the rules themselves? The discussion presented in this review aims to address some problems by highlighting scientific aspects, but also by summarizing psychological aspects that can simplify the scientific basis, not in terms of validity, but of quantity.

To change points of view and break the rules of the game, two fundamental tools should be used: Pop culture and irony. So, a pop culture aspect is considered first.

The new millennium began with the strange idea of replacing the internal combustion engine (ICE), characterized by an extremely high energy reserve in a tank, with an electric motor, powered by limited-size batteries. This choice has already put the comics industry in crisis, which has not yet found a single onomatopoeia for the noise of the cars. "Brooomm,"

"Drooow," "Vroom," and "Roammm" are the standard noises, but now we will have to find something more significant than "Zzzz!".

The matter is serious. To solve the problem, the iconic German manufacturer of Bayerische Motoren Werke (BMW) performance machines asked for the help of one of the greatest modern composers of movie soundtracks, Hans Florian Zimmer, to create a sound of electric cars worthy of the ICE sisters [1].

In order to face the issue of the silent arrival of vehicles and pedestrian protection, the European Parliament delegated a commission [2]. The result was a noise device to be adopted, called the "Audible Vehicle Alert System," which has already triggered a war of noise. In 2021, Maserati, the luxury brand of the Fiat Chrysler Automobiles (FCA) group, will introduce its first electric car, the Granturismo, in order to study the "soundtrack of the Trident" of its electric vehicle (EV), and is developing an iconic and distinctive sound at its Innovation Lab development center in Modena. The same philosophy has been adopted by Porsche in its Taycan luxury electric sports car. The buyer can, in fact, decide to add the Electric Sport Sound to his car, an optional item with a cost of EUR 500, which adds a real soundtrack, both inside and outside the car. The Jaguar Land Rover has opted for a more ordinary sound, which, despite using expert sound engineers for its I-Pace SUV, comprises a simple acoustic warning. The debate is therefore quite open: Must the sound of the car of the upcoming era have its roots in the past or must it make us listen to the sound of future?

A scientific aspect lies in the fact that the European Commission requires the use of an acoustic alarm, but the rules of the game imply that it is more important to perceive the performance of the vehicle than the vehicle itself. This is the first paradox that we meet when approaching the world of electric cars.

The great paradox that every nation has faced or is about to face is that of the "chicken or the egg" [3]. The question is whether the market for EVs in a given region can develop with or without the prior creation of a dense electric recharging network. To address this problem, various scientific works have been carried out which have contributed to generating a profile of the first adopters of this technology. So, the various sections of this work are dedicated to the adoption phase (user definition, fears of performance, the search for the most fertile market niches...). However, before adoption, there are common phases, such the perception phase and the use phase, or "how others perceive the fact that I am driving an ecological vehicle?" This aspect is addressed symbolically in the following paragraph. Therefore, some sections are dedicated to perception regarding the vehicle (problem of marriage or cohabitation?), future use (silver vehicles), and the notions of status symbol and gender attitudes (Viking man).

If the reader is looking for simple answers, the author does not recommend reading the subsequent sections, as few paradoxes are solved.

Section 2 addresses the problem of finding early adopters who will help in the diffusion of the innovation. The search for a market niche is a common theme for several sections, so it is addressed first. The concept of leaders and followers is perhaps a more efficient lever than the analytical evaluation of the total cost of ownership. "How do others see me?" is more important than "How much do I save in 10 years?" Section 3 deals with the chicken and egg paradox: Which came first, the electric vehicle or the charging station? This survey presents some psychological aspects which have a strong impact. An excellent excuse to stop innovation is to use the rhetoric of reaction, stating that the system is not ready for the adoption of new technology. On the one hand, it shows how, on characteristics of exclusivity, it could develop an autopoietic system, which could also obtain a certificate for being eco-friendly. Section 4 challenges a psychological fear: Range anxiety. The human being collects various anxieties, or inadequacies, in working relationships. His carriage has always distinguished him from knights and foot soldiers. Is he ready to add another anxiety related to the carriage, his status symbol? Section 5 attempts to dispel the fear of the explosion of electric vehicles. In an age of digital information, the sound of a falling tree is louder than that of a growing forest, so is one vehicle catching fire an original sin for all future vehicles? Section 6 addresses the problem of gender attitudes and how other

view the owners of electric vehicles. Commonplace notions are hard to dispel, but can the renaissance of electric vehicles change the rules of the game? Section 7 deals with the upcoming era of automated vehicles and the ways in which they are perceived. There is only one certainty regarding the vehicles of the future: They will still have wheels, for a long time. Are human beings willing to trade the pleasure of driving for greater comfort and widespread well-being? Finally, Section 9 describes changes of attitude after the trial of an electric vehicle and concludes the paper.

## 2. Finding Early Adopters

In 1962, Everett Rogers published his "Diffusion of Innovations" [4], defining five categories/customer types, shown in Figure 1:

1.  Innovator. They are a small group of people exploring new ideas and technologies, also bored by the previous ones. This group includes "gadget fetishists". In an online marketing context, there are a lot of specialist blogs and media sites to engage such individuals.
2.  Early Adopters. Considered to be "Opinion Leaders" who may share positive testimonials about new products and services, they can show the efficiency of EVs.
3.  Early Majority. These are "Followers" who will read reviews by earlier adopters about new products before purchasing.
4.  Late Majority. To generalize, these are sceptics who are not keen on change and will only adopt a new product or service if there is a strong feeling of being left behind or missing out. They should buy an EV, but are not enthusiastic.
5.  Laggards. The descriptor says it all! Typically, they prefer traditional ICE and will adopt new EVs when there are no alternatives. Laggards are convinced of machinations and have their own ideas on everything, often supported by pseudo-scientific reasoning.

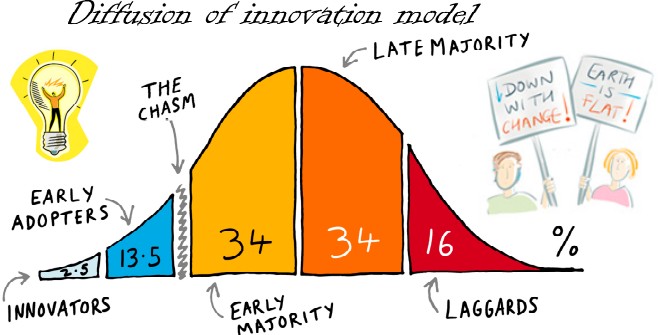

**Figure 1.** Diffusion innovation model of Rogers.

For the diffusion of new technology, it is necessary to surpass the so called "Chasm", so various efforts have been dedicated to understanding which incentive policy can help in this phase.

In this section, we devote our attention to defining the profile of Early Adopters. Most surveys are carried out by providing a questionnaire to demographics which may be interested in the novel technology and analyzing the answers with clustering algorithms, to highlight, if possible, the category that best responds to the figure of Early Adopters. It is curious to note how similar surveys define different profiles. Some investigations have a broad spectrum, encompassing entire nations, but others use a narrow band, as in the first case we will address, which was limited to only one city.

It is well known that there are barriers to purchasing an EV that fall within the socio-economic classification. In [5], a clustering algorithm was generated based on user characteristics such as age, income, car ownership, home ownership, socio-economic status and education. Nearly 60% of the zones in the city of Birmingham fitted the profile for

alternative fuel cars. The areas where the poorest people live were located in the central part of the town, as is common to many cities.

By following Rogers's model, the early adopters desire to be the first to own alternative fuel vehicles (AFV) and want to see themselves as models of society. The early majority consists of those who will spend longer time deliberating over buying the AFV, waiting the response of early adopters. In contrast, late majority adopters are somewhat more careful and skeptical. Finally, laggards remain linked to traditional vehicles, do not have the resources to buy an AFV, or lack knowledge and understanding of such vehicles. The aforementioned research concentrated on the following items: (1) age; (2) home ownership; (3) home detached or semidetached; (4) drive to work; (5) owning two cars; (6) income level and socio-economic status; (7) education. The survey identified the profile of early adopters in Birmingham: people with greater affluence, higher rates of car ownership, higher income, and higher rates of home ownership (points 2, 3, 5 and 6).

A national survey in USA identified early adopters as young, very high-income individuals, house owners, with the perception that EVs are green and clean, who own their own car and drive 100 miles per week [6]. In contrast, "nonadopters" have low incomes and do not have their own garage, creating a challenge for safe and secure home charging. So, the lack of infrastructure reveals the problem of "the chicken and the egg", which will be addressed in the next section.

Again in USA, another survey [7] defined early adopters as younger to middle aged, having a Bachelor's or higher degree, anticipating higher gasoline bills in the following years, environmentally minded, having a garage or a space to charge at home, and inclined to buy new goods that come on the market.

Instead, an investigation [8] involving buyers of the Toyota Prius in UK showed that men aged 50 and over were most likely to purchase such a vehicle.

In [9], a comparison was made of the behavior of early adopters in China and Korea. The analysis took three factors into account: functional, symbolic and experimental motives. It finally drew profiles for the two countries. Both embraced small-sized EVs principally. The preference among Chinese people was found to be for first-time purchases, showing a higher level of environmental care higher compared to that of the Korean early adopters. Positive experiences in electric taxis were also a factor among Chinese people, in contrast to Koreans. A very important psychological factor is "how others see me" or "how one thinks about someone driving an EV"; these questions may be answered as follows: "An EV differentiates me from others", "an EV suits my lifestyle", "an EV makes me seem environmentally friendly", "an EV shows that I am technologically advanced", or "an EV shows that I am a socially responsible". Chinese early adopters placed the highest degree of importance on environmental factors, whereas for Korean early adopters, it was the economic factors. For the Chinese early majority, economic reasons only placed third, while for Korean early adopters, environmental factors placed third. The demographic profile of Chinese responders showed: female (51%), 31–40 years of age (64%), bachelor's degree (71.1%), a monthly income of 1500–3100$ (49.7%); and for Korea, male (82.5%), 31–40 years of age (49.7%), bachelor degree (73.4%), and a monthly income of 3101–6000$ (37.9%) or 1500–3000$ (36.7%).

A further study confirmed the profile of Chinese first adopters: female (57.5%), age 26–35 (48.7%), college education (49.8%), and income 2001–4000$ (30.7%) [10].

Also, early adopters in Switzerland were analyzed [11]. The analysis described the barriers for diffusion, noting that only one family out of nine would buy an EV as their primary car, six out of nine would buy an EV as second (or third!) family car, and five out of nine would not buy an EV, preferring a HEV.

The authors of [12] presented an analysis of the attitudes of UK drivers regarding the forthcoming ban on the sale of internal combustion engine vehicles.

In order to look for common behavior among early adopters, perhaps economic parameters are a common denominator, as both age and gender fail. Very often, research carried out in the same country has shown contrasting data.

Last but not least, an analysis was performed in the northern countries of Europe. Norway surpasses other countries in registrations (see Figure 2) [13,14]. The European Environment Agency (EEA) [15] reported that 22.5% of all new cars sold in Norway in 2015 were electric. In [16], an analysis found that respondents were equal in percentage for gender and age: <25 (18.2%), 25–34 (18.0%), 35–44 (18.1%), 45–54 (19.2%), 55–64 (15.6%), 65+ (10.9%).

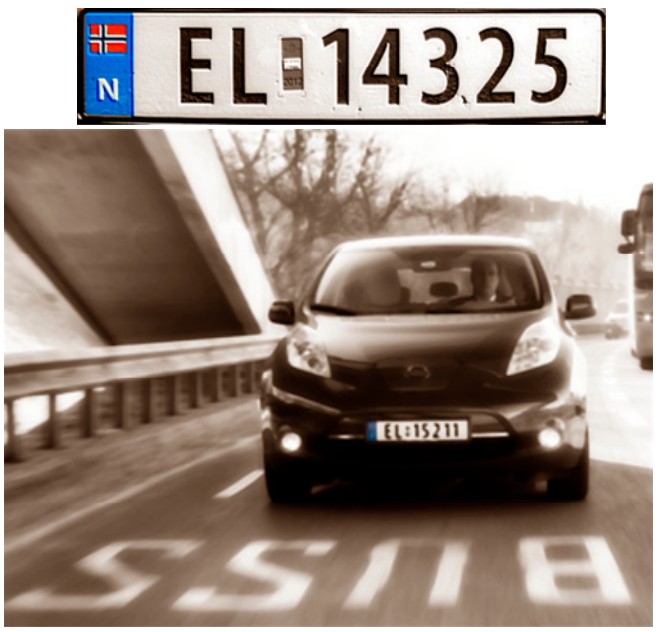

**Figure 2.** Proud Norwegian license plate (EL indicates electric vehicle).

In conclusion, it can be said that the ideal user may not exist, as the sales data of vehicles in the regions with the highest registration rates do not show trends in terms of age, sex or education; rather, perhaps we have passed from early adopters to the early majority, thus making the demographic transversal. The first study that was considered had the merit of stating that the only variable that was not confirmed was higher education, meaning that it is not necessary to study to own an EV. Fortunately, subsequent studies have shown that higher education is often correlated with environmental sensitivity. However, studies have shown a tendency to own EV as a second vehicle, which is possible only for wealthy persons. The same studies questioned the gender identity of the electric vehicle, and will be discussed in the section, "Viking men paradox".

A question arises: why did Stockholm not become the city with the greatest number of EVs, despite the existence of all the favorable conditions to incentivize early adopters [17]? The European Commission's Innovation Scoreboard placed Sweden as the first among EU member states in 2013 and 2020 [18,19], so why does Stockholm lag behind Oslo and Copenhagen? In [17], different hypotheses are presented, principally divided into three shortcomings with different discretization levels, from niche initiatives to national directives. (1) Niche initiatives: very few home-grown niche initiatives occur in Stockholm, resulting in incomplete awareness, experience and knowledge of battery EV (BEV). Local stores did not perform demonstrations. (2) The was a lack of regime patchworking initiatives; rather, initiatives mainly supported alternative fuels and PHEVs, opposing BEVs. For the cognitive dimension, it is important to explain the differences between Stockholm and Oslo, in which a greater prominence of BEVs is highlighted, since EVs are easily recognizable by their license plates [16]. (3) A lack of wide scenario policies: policy directions, visions and economic incentives confuse users, who do not know if private vehicles or collective ones will be supported in the coming years. The solution? As many

studies have shown, the experience of driving a BEV is unforgettable; this will be discussed in Section 8.

## 3. Chicken or Egg Paradox

The main paradox that every nation has faced or is about to face in encouraging the diffusion of EVs is the lack of charging infrastructure. Private and public infrastructure is not established before the initial circulation of EVs. But users are waiting for the construction of a net of charging station before purchasing an EV. It is a "chicken or egg" paradox.

The paradox is already mentioned by ancient Greek philosophers such as Aristotle and Plutarch. But the first to formulates it in the way we know it today was Ambrogio Teodosio Macrobio, in his work *Saturnalia*: "Ovumne prius extiterit an gallina? [3]". The question is whether the market for electric vehicles in a given region can develop without the prior creation of a dense electric recharging network. Like with the chicken and the egg, neither can exist in the absence of the other.

It is not trivial to tackle this problem, as it lends itself to a cross-sectional analysis, whether it addresses early adopters, range anxiety, or other grievances towards EVs. The common man is still inclined to think that the EVs will have to be recharged like ICEs, that is, at a service station that can guarantee autonomy for hundreds of kilometers, all within a few minutes, thus feeding the idea of battery swapping. This point of view falls into the hegemonic present. Battery swapping, in some cases, could expose the batteries to the risk of explosion following an impact accident since the batteries should be placed not in the inner part of vehicle, and since the electrical-thermal control action risks being less efficient on a system that is not necessarily exclusively projected for the electric vehicle in question. This will be discussed in section entitled "my cousin told me EVs explode".

Let us consider a scientific point of view. Viola et al. in [20] described the performance of different cities in Italy to address this issue. Figure 3 shows a chart in which, on the abscissa, the number of initial users (early adopters) per coverage station was divided by the size of the city in question, while on the ordinate, the initial user density was weighted by the number of the charging infrastructure. In this chart, we find three levels. In the first, the cities that performed best surpassed the weight of large number of charging stations and density. Level 2 shows the median attitude of different cities. Level 3 shows the performance of laggard cities. This reveals a direct correlation between EVs and charging stations. But in view of research on early adopters conducted applying the aforementioned parameters, the cities present in level 1 matched at least two of the four items presented in a previously survey: greater affluence, higher car ownership, higher income, higher home ownership.

This graph allows us to understand which cities were most favorable to EVs, and can be used to predict the adoption of auxiliary systems for the same purpose, such as those on the roadside which are necessary for the coordination of autonomous driving vehicles. It is necessary to recall that in order to achieve the greenhouse gas mitigation targets, plug-in electric vehicles (PEV), both BEV and PHEV, have to be powered with renewable energy. Cities in level 1 of Figure 3 require a better quality of air [20].

Different reviews faced the "chicken and egg" problem and highlighted the different trends in various countries [21,22], but also for gender and age [23].

By following the approach applied in [21], we can distinguish between demand of charging infrastructure and need; these two categories differ as subjective and objective necessities. The demand is indicated by the empirical charging behavior of users (generally inside the city), while need is estimated based on the charging required to travel certain distances (intercities). Charging infrastructure needs are affected by subjective parameters such as comfort and range anxiety (addressed in the following section). Once again, it is necessary to make a distinction between habits of the hegemonic present and real perception. High-power, fast charging stations should be used to ensure continuity corridors, not as a substitute for a full of tank for the sake of convenience.

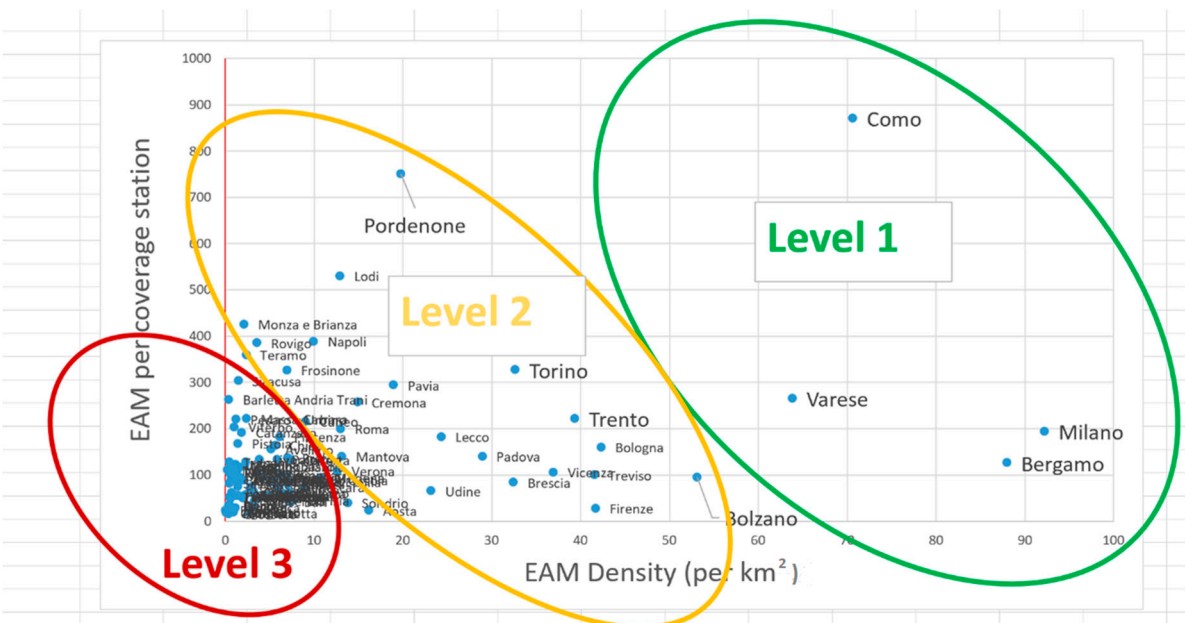

**Figure 3.** The chicken and egg paradox: performance of different Italian cities by considering the number of early adopters weighted by number of charging station and dimension of city.

The importance of a strong and pervasive charging infrastructure is fundamental to attract the early majority.

In [21,22], it was shown that 50–80% of all charging events occur at home, second furthermost significant charging place is at work, where 15–25% of PEVs find their energy; less than 10% of all charging actions happen at the remaining sites. In the competition between EVs and Charging stations, Plug-in EVs born first, the numbers show it. By returning to the 10%, although the use of this structure may seem reduced, its action is mainly to convince people of the existence of the alternative to the use of ICEs.

In [21,22], it was shown that 50–80% of all charging events occur at home, while 15–25% occur at work; less than 10% of all charging actions happen elsewhere. Although the cases associated with the use of charging stations may seem low, their presence serves to convince people of the existence of an alternative to ICEs. Micro- and mid- sized hybrid vehicles are not the harbingers of a revolution, PHEV are often not connected to recharge, so the limited number of users, i.e., those with detached houses and garage owners, are not able to support the passage to the early majority. To inform of the presence of an ecological alternative to vehicles that use fossil fuels, battery vehicles are necessary that avoiding autopoietic errors, e.g., domestic recharging in detached houses owned by wealthy people, or, as in the case of Stockholm, via underground charging stations, which may be perceived by women as dangerous areas (see the section entitled Gender Attitudes).

So, the 10% is more important than the 90%, and charging stations should born first!

In order to perform a better analysis of the role of charging infrastructure, attention is now focused on light duty EVs, considering three types of public charging infrastructure: (1) near homes as a substitute of private charging; (2) charging near points-of-interest (grocery stores, cinemas, etc.); (3) fast charging stations, to ensure long travel corridors (typically DC stations).

Near home charging is required for users who are not owners of detached houses with a garage. By considering large cities with high number of inhabitants per square kilometers (or a high number of light vehicles), a widespread public charging network is needed. A parameter to establish the efficiency of this charging network is the vehicle-to-refueling index (VRI), i.e., the number of PEVs per charging point [21]. A high VRI indicates either a developed PEV market or underdeveloped infrastructure; a low VRI indicates either a less developed PEV market or a large number of public charging stations. Sweden, the US and

Norway show high VRIs, ranging between 12–19 PEV [21]. For Norway, this is partly due to the incentive of the free use of ferries, car parks and public charging stations [14], which has the effect of crowding charging stations. A low VRI was found for The Netherlands (4 PEVs per charge point), which indicates a diffuse public charging infrastructure. In order to combine the VRI index with the graph shown in Figure 3, it can be understood that a high VRI conforms to the high abscissa; a low VRI could describe very large cities in which charging stations are easy to find.

An interesting aspect is that for VRIs of a few units, a station for ICEVs has two thousand users, so the presence of a PEV at the ICEV station could be a good advertisement, but it would be like asking a chain of junk food restaurants to support vegetarian and healthy food.

In order to better define the VRI, the following figure, obtained by recent data [24], should be discussed. On Figure 4, Norway is not represented correctly; the country has 905 charging stations for every 100 km of highway. This number is affected by the fear of being stuck in an isolated region during winter, as discussed below. Similar VRIs for the abscissa as those of The Netherlands and Italy, or France and Germany, show similar performance, but in The Netherlands, the low number is due to a highly developed charging station network, whereas in Italy, this is not the case. This can be seen by considering the charging stations that address actual need [21].

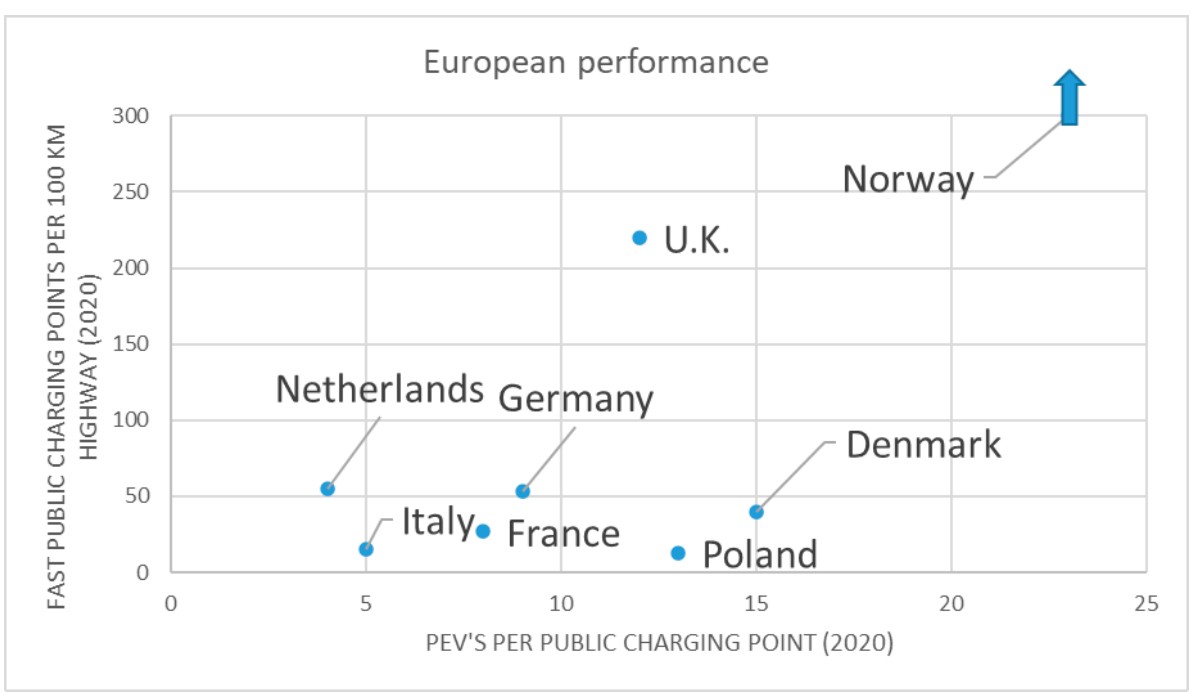

**Figure 4.** The chicken and egg paradox: vehicle-to-refueling index VRI for European countries.

An interesting paradigm regarding charging at points-of-interest was presented in [25]: a recharge area on a university campus was designed according to users' attitudes, thus maximizing the energy produced by a photovoltaic system. A similar study was undertaken by [22] for home and public recharging (grocery stores, shopping malls, and in parking lots). In this way, an ever-greater vision for PEV can be guaranteed.

Figure 5 represents the level of interest of EV users for the placement of charging stations [26].

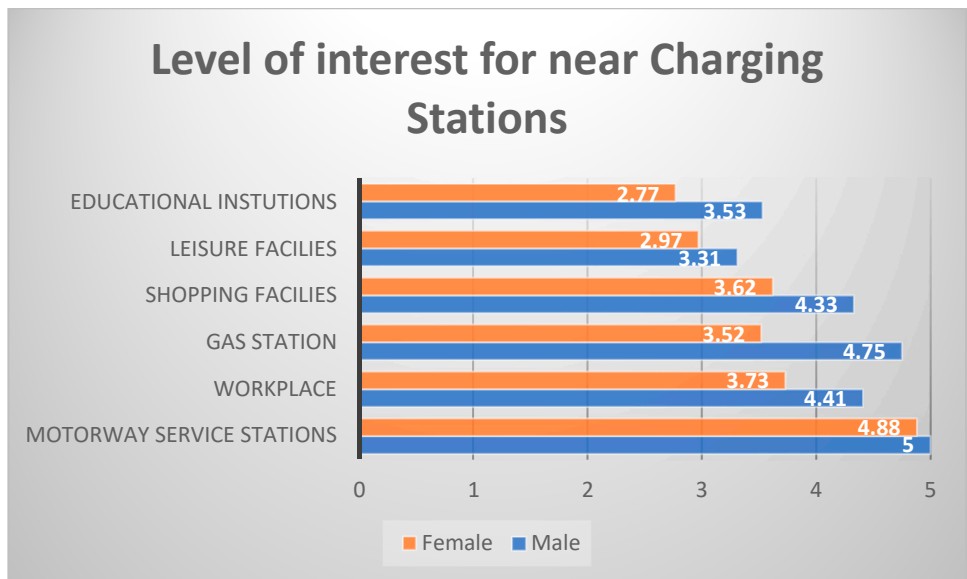

**Figure 5.** The chicken and egg paradox: interest level among users for vehicle refueling. Women beat men in terms of the use of charging stations at educational institutions!

Previously, we said that PEVs were born first. This, however, is not the goal, since the result applies to early adopters, leading to an autopoietic system, i.e., a system capable of reproducing and maintaining itself without external contact, completely autonomous from the external system and not interacting with it. If the owner of an electric vehicle owns a detached house and a photovoltaic system, he may not be interested in public recharging facilities, and will therefore not contribute to stimulating the transition from early adopters to the early majority, as was the case of Stockholm with underground stations [16,17]. Therefore, the remaining 10% leads to the creation of an allopoietic system. Figure 6 shows an allopoietic system in which there are interations between different people. In Italy, there is the tradition of lowering a basket, a "panaro", from the upper floors to have bread delivered. Even if, from a safety point of view, the panaro is not a safe method [27], it encourages the spread of electric vehicles.

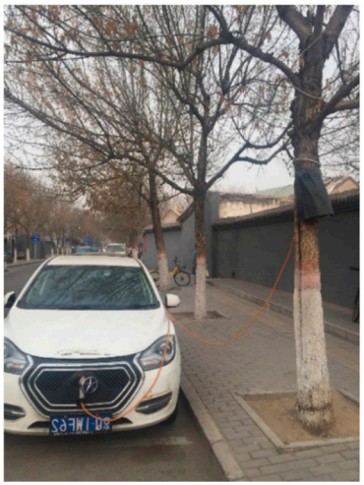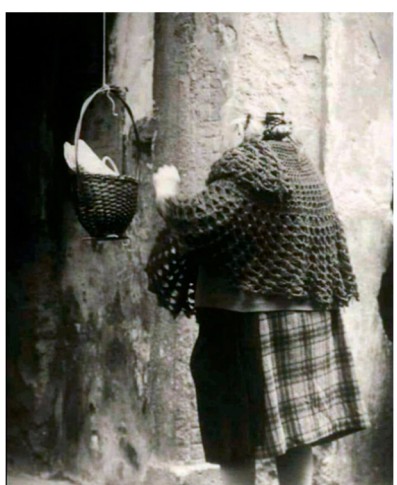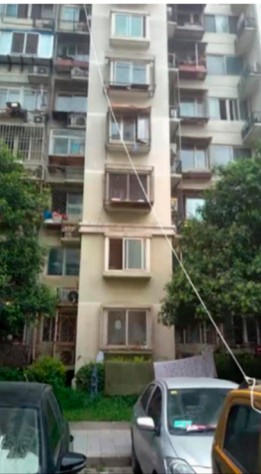

**Figure 6.** The chicken and egg paradox: solving the absence of charging station problem, the "panaro" solution, also named "fly line" in the technical report [27].

From these arguments, it can be estimated that PEVs were born before public charging infrastructure. On the other hand, the first speeding infraction in the U.S. was committed

by a New York city taxi driver in an EV on 20 May 1899, when there was still no idea of public charging stations [28].

## 4. Range Anxiety

Performance anxiety in the collective imagination is the main psychological factor that constitutes a barrier to the spread of EVs. The scientific articles that address the problem of "Range anxiety" are many [26,29–36], and we believe that many of these have some of the most hilarious and amusing titles a researcher can ever find: "Does range matter?...","Fast-charging station here, please!...", "Inaccuracy versus volatility—Which is the lesser evil in battery electric vehicles?", "Running on empty . . . ", to cite some of them.

By range anxiety we mean the anxiety of not succeeding, not reaching the goal, an anxiety of performance. This is one of the main obstacles to the spread and social acceptance of BEVs. The battery, seen as a weak element, obscures all of the many peculiarities that ought to establish the BEV as a winner (reduced energy consumption, very low number of moving parts, reduced maintenance, no direct emissions, etc.).

Before describing the various types of users and their anxieties, it would be advisable to investigate the history of this phenomenon in order to understand what an ICEV represents in the collective imagination.

By following the research of [37], the changes that caused producers and customers to abandon bicycles, horses, EVs, cable cars, trolleys, and trains for ICE powered vehicles can be seen to have occurred in the fifty years from 1890 to 1940. From 1895 to 1910, EVs were more common than ICEVs in most regions of the USA and Europe. This could be considered the golden period for EVs. The decline began in early-1910.

The Ford Motor Company opened its Highland Park Plant in 1910, and subsequently implemented a moving assembly line there in 1913, thereby reducing the cost of its model T. By the 1930s, the popularity of EVs had all but evaporated. Looking for a simple reasoning, many engineers and technical experts explain the disappearance of EVs and the rise of the ICEVs as solely a technical matter. They note that EVs suffered insurmountable technical handicaps, among which expensive batteries with limited cycle lives and long recharging times, poor acceleration, and limited range figure prominently.

In 1890 the primary means of transport were horses and horse-drawn carriage. In a day, a horse team might cover up to twenty, miles, with an average step of 3–5 miles per hour. ICEVs were limited by a lack of uniform spare parts. As such, EVs became a popular choice from 1900 to 1910. Commercial workers saw numerous benefits to using EVs. Commodity suppliers of coal, ice, and beer (which we would call energy carriers) predominately used EVs to distribute goods to their customers. Electric trucks had an operating range that was greater than that of a horse wagon, but less than that of an ICEV. In 1910, ICEVs started to overcome EVs due to four interconnected phenomena: technical, economic, political, and socio-cultural issues. Technical factors: ICEVs went from having three-horsepower, noisy, and unreliable motors to having 30-horsepower, efficient motors in 1905, and spare parts started to become more widely available. Economic factors: The Ford model N was the cheapest car on the market in 1905 ($500). Political errors: Electric companies lacked the momentum, mainly focusing their attention on large-scale rural electrification projects and building alliances with electric appliance manufacturers, paving the way for oil companies and gasoline automakers. Socio-cultural factors: the coup de grace to the dominance of EVs was dealt by a set of socio-cultural aspects; EVs were associated with conservatism and femininity [36], as they were easy to operate, and lady drivers especially liked their cleanliness and simplicity; furthermore, their lack of power was mocked by men (this aspect will be taken up again in the Gender Attitudes section). At the same time, the "end of the frontier" phenomenon occurred, and more and more complex industrial and capitalist realities arose; the United States was becoming too European. In a such scenario, the limited range of EVs cooled the desire to undertake journeys into the wilderness. ICEVs fulfilled the wish to experience the frontier. A longer range also allowed made longer trips possible. For example, between 1915 and 1924,

Henry Ford, Thomas Edison, Harvey Firestone, and John Burroughs, calling themselves the Vagabonds, embarked on a series of summer camping trips. Social and cultural forces played a fundamental role in transportation decisions, and ICEVs made it possible to build a connection between wilderness and civilization, while BEVs suffered, for the first time, from range anxiety syndrome.

At the same time, European countries experienced the vicissitudes of the great war. In terms of logistics, ICE trucks were very useful at the front, thereby accelerating the decline of means of transport with horses and electric motors.

Range anxiety is defined as the psychological anxiety a consumer experiences in response to the limited range of an electric vehicle [31]. In [33], the authors defined three range levels: competent, performant, and comfortable. The first and second are based on the technical knowledge of the user on his vehicle and driving skills; competence may be associated with self-regulated learning. Performance also employs subjective subscores due to the idea of range starting with a fully charged vehicle and the possibility of reaching new goals. Finally, the "comfortable" range is a psychological one. The relationships among the range levels yield user information. A higher "comfortable" range could allow drivers to reduce their efforts to increase their available range for everyday driving.

In order to explore range anxiety, the authors of [31] extended the technical and phycological motivation of anxiety to include a third category: intransigence, as derived through Hirschman's rhetoric of reaction [38].

Technical anxiety finds a numerical measure, it happens "when the distance to be traveled is greater than the vehicle's range and a the simplest response includes an investment in charging infrastructures or in increase of batteries capacity"; a phycological anxiety arises "the distance to be traveled is below to the vehicle's range, but users irrationally are worried about the possibility to finish the charge", and a simple response for this attitude may be found in driving experience and education (see the section: Cohabitation or Marriage). Rhetorical anxiety is very different, and the previous explanations fail to address it. The rhetoric reaction "masks" a deeper insecurity, thereby creating a barrier to the purchase and use of EVs.

In his The Rhetoric of Reaction: Perversity, Futility, Jeopardy [38], Hirschman analyzes the rhetoric of "intransigence" or conservatism in opposition to innovation. On the one hand, conservatives believe that reformism generates perverse effects (i.e., the ultimate effect of a reform is the exact opposite to what it intended to achieve), futile (the final effect of a reform is nil, not modifying the pre-existing situation) and/or dangerous (the final effect of a reform is harmful, in the sense that it involves a reduction in general well-being). So the innovation of EVs will create a paradox in terms of generating electric energy from dirty energy sources (perversity); expensive EVs are not changing this world (futility); overpriced EVs and policies exclude all but the wealthy (jeopardy). For range anxiety, the authors of [31] propose that EVs lack the range required to safely make trips, and that persons fear being stuck in isolated settings (jeopardy). The latest fear is so felt in Norway that it has created a very dense motorway charging network (the higher number of charging stations on Figure 4).

In order to measure anxiety, we may consider Figure 7, in which the majority of daily range need is shown to be 0–80 km, and the mean daily driving range is 72 km, with a median about 48 km. The orange line shows that trips of 160 km or more in a single day typically occur only 24 times per year [30].

Figure 8 summarizes the work presented in [34]. It indicates that during travel time, SoC decreases and anxiety appears after a comfortable range threshold, indicated in [33] as buffer.

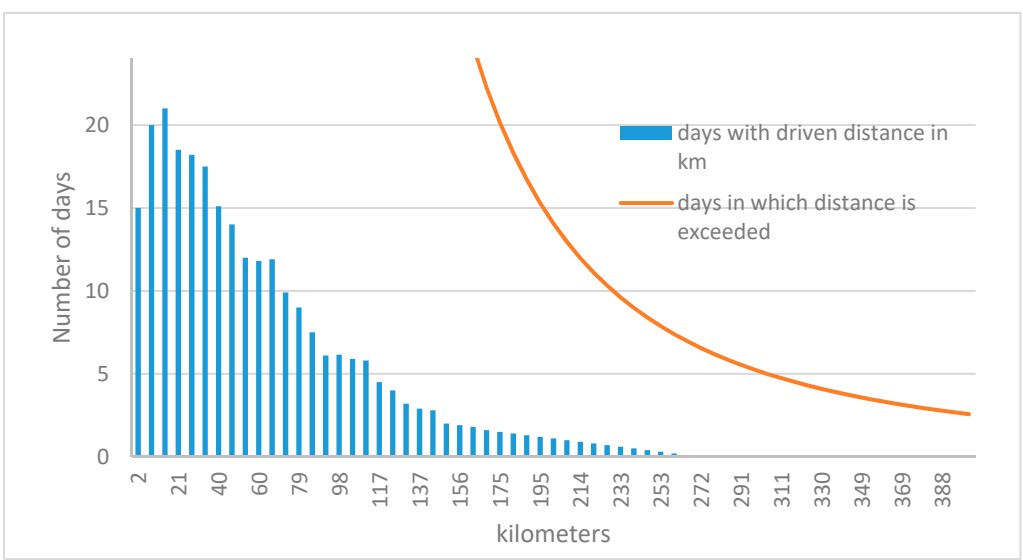

**Figure 7.** Average daily distance distribution [30]. Blue bars represent the number of days on which a given distance is covered, while the orange line represents the number of days on which that distance is exceeded.

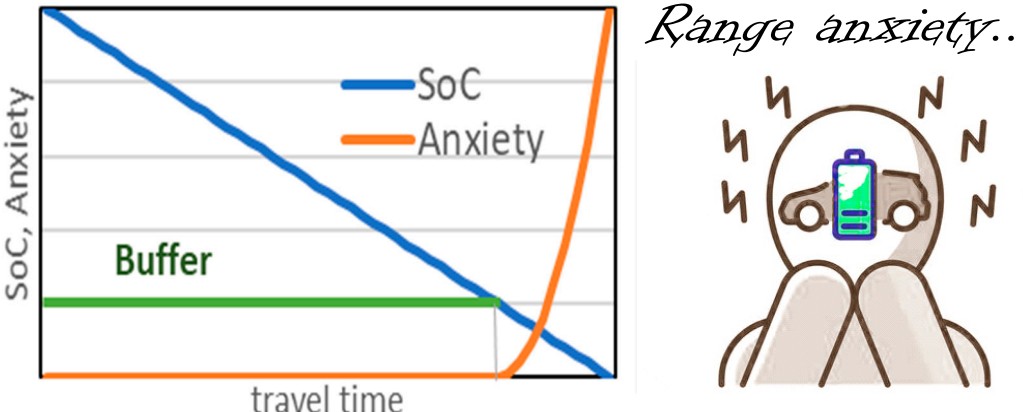

**Figure 8.** Profile of SoC and driver range anxiety.

In order to introduce an extremely simplified reasoning, we consider driving an e-golf, with 300 km of autonomy according to the manufacturer [39]. Considering a buffer of 20% as a comfort level, the EV would have 240 km of autonomy if starting with a full SoC. Based on Figure 6, such a journey would typically be undertaken eight times per a year. If we then consider future improvements in battery packs, allowing them to achieve autonomies of 400 km in the next three years, the days of anxiety would be halved to four. The example of a family car from the perspective of average vehicle use showed the analytical value of the emerging anxiety about the "need" for charging stations (see previous paragraph). In places like Norway, due to the fear of running out of energy in the middle of a storm, many stations have been built on long-distance corridors.

Something that happens for four or eight days every year, such as a cold, should not constitute a serious fear, especially if it prevents the use of elements that contribute to reducing pollutants, that also favor the spread of colds.

In conclusion, range anxiety may be said to be more psychological or rhetorical, until we discover within ourselves the desire to reach the frontier. We can conclude with Aesop's famous fable about the fox and the grapes. The fox, not being able to reach the bunch of grapes, declared that it was unripe. The user who, due to rhetoric of reaction, does not want to switch to an EV, will always say that his wishes are out of reach.

## 5. My Cousin Told Me EVs Explode...

An often-distorted view of EVs is that they can catch fire and explode, jeopardizing the safety of the user, and thus, justifying the rejection of this new technology. This rejection was previously faced with the rhetoric of the reaction to innovation. There are many documented cases in the technical literature of electrical fires [40–48], some of which were even difficult to extinguish, but generalizing the concept to all EVs is incorrect and dangerous.

To understand the risk, it is necessary to consider the elementary lithium-ion cell, made up of an anode, cathode and electrolyte in a solid–electrolyte interface (SEI) [41]. Also, the separator is of fundamental importance, as it guarantees the separation of the electrodes. This element is put at risk by mechanical, electrical and thermal accidents. The breakage of the separator places the two charged elements directly in contact with each other, and the resulting chemical reaction requires almost no external contribution, so is unstoppable.

The tearing of the separator following mechanical impact has been studied and addressed in different studies. Following mechanical abuse (e.g., after an accident), the battery pack can undergo a deformation or even a penetration by external objects; in this way, the function of the separator is no longer guaranteed, and internal short circuits (ISCs) can be triggered, compromising the cell and the neighboring ones. Battery packs are therefore positioned in more internal places, making them less accessible, so improved safety precludes battery swapping, the solution proposed by those who transport the present into the future (hegemonic present).

Mechanical abuse was found in the case studied in [43]. The National Highway Traffic Safety Administration (NHTSA) in 2011 opened a defect investigation into a Chevrolet Volt. NHTSA had performed a side-impact test on the Volt, and then parked it outside, and three weeks later, the PHEV caught fire. It was necessary to determine if the situation could happen again. Similar tests were reproduced and another Volt caught fire a week after an accident. In the first case, the battery pack was damaged an lost its coolant fluid, so the temperature progressively increased. That led the NHTSA to consider a ruling requiring hybrid and electric-car batteries to be drained after an accident. However, before establishing that road accidents can lead to a fire risk even after weeks, it is right to refer to other studies. In research carried out by Dekra [44], an expert in road safety, in collaboration with the University Hospital of Gottingen in Germany, similar crash tests were carried out on electric cars. Despite the severe impact to which the cars were subjected, the resulting severely damaged batteries did not catch fire, as the high voltage system was effectively shut down by the safety systems during the accident. EVs were thrown against a pole, simulating a frontal crash at 84 km/h and a side crash at 75 km/h. With the second type of accident, the driver would very likely not have survived, be it in an electric or a conventional cars. The potential buyer may no longer be willing to purchase an EV or ICEV after seeing how the vehicle crumpled in the accident; he would prefer not to leave the house anymore, given the severity of the accident. See Figures 9 and 10.

The second type of abuse is electrical. Within this category, we can distinguish between overcharge and over-discharge abuses. The failure of the battery management system to stop the charging process before reaching the upper voltage limit is the usual cause of overcharge abuse. During charging, the increase in voltage is accompanied by a temperature increase. After 100% SoC, there is a reduction of lithium from the cathode, generating a sudden increase in cell temperature [42] depending on the chemistry of the cathode (varying from 100 to 200% of SoC), potentially leading to the thermal runaway phenomenon. In addition, during overcharging, the lithium plating on the anode can create a dendrite path, linking the anode to the separator, and thereby generating internal short circuits. The overdischarge process is similar: during discharging, the stability of the cell depends upon the solid–electrolyte interface (SEI). If the SEI is decomposed, a copper dissolution can create a short circuit, breaking the separator.

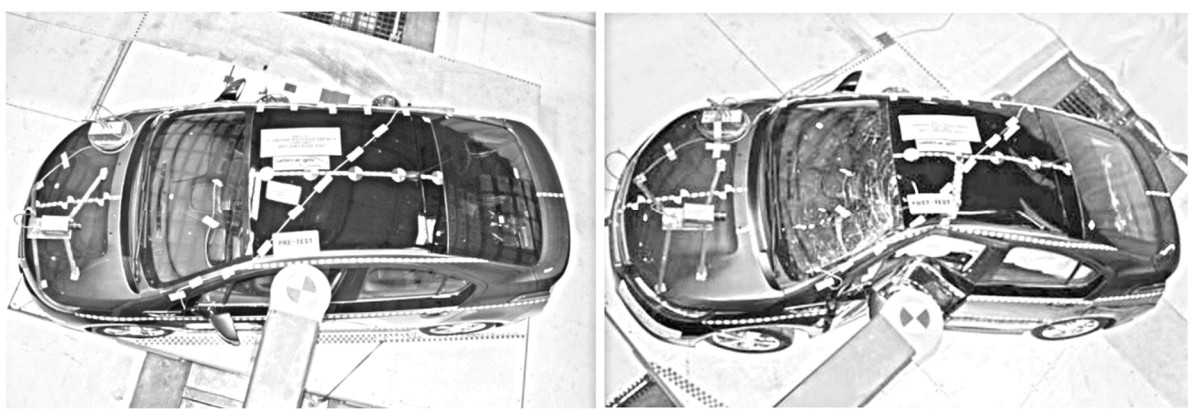

**Figure 9.** Chevrolet Volt NCAP pole test, pre- and post-test [42].

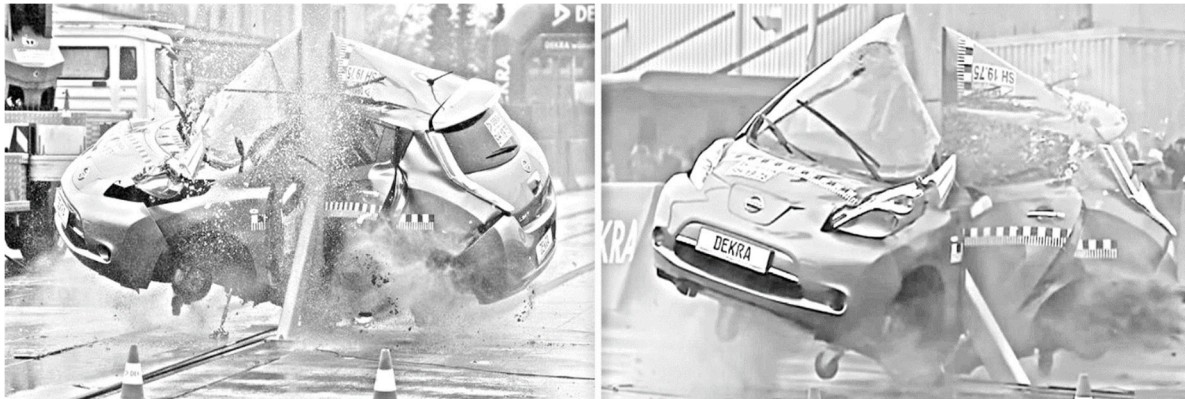

**Figure 10.** Nissan Leaf pole test by Dekra [44].

Examples of electrical abuse can be found in a report on aircraft malfunctions [45–47]. On 7 January 2013, an incident occurred involving a Japan Airlines Boeing 787-8, JA8297, which was parked at a gate at General Edward Lawrence Logan International Airport, Boston, Massachusetts. Maintenance personnel observed smoke coming from the lid of the auxiliary power unit battery case, as well as a fire with two distinct flames at the electrical connector on the front of the case [45]. No passengers or crewmembers were aboard the airplane at the time, and none of the maintenance or cleaning personnel aboard the airplane was injured. The National Transportation Safety Board determined that the probable cause of this incident was an internal short circuit within a cell of the auxiliary power unit (APU) lithium-ion battery, which led to thermal runaway that cascaded to adjacent cells, resulting in the release of smoke and fire. On 16 January 2013 [46], nine days after the previous incident, a Boeing 787-8, operated by All Nippon Airways Co., LTD., took off from Yamaguchi Ube Airport for Tokyo international Airport at 08:11 local time. During its ascent over Shikoku Island, a message of battery failure came at 08:27, accompanied by an unusual smell in the cockpit. The airplane diverted to Takamatsu Airport and landed there at 08:47. An emergency evacuation was executed using slides on the T4 taxiway at 08:49. Four passengers out of 137 suffered minor injuries during the evacuation. Although the main battery was damaged, it did not lead to a fire. An internal short circuit was the cause. Another, similar incident occurred at the Narita International Airport on 14 January, 2014 [47]. While preparing for the departure of a JAL 787 aircraft from Narita airport from parking spot 72, a maintenance technician in the cockpit noticed white smoke coming from under the fuselage. The technician went outside immediately but did not see any smoke. Upon returning to the cockpit, the technician noticed messages showing that the main battery and its charger had anomalies. The voltage of the main battery was 27 V. There was no record that any messages for abnormal battery voltages had been displayed during the previous flight, even though the battery should have a maximum voltage of 32 V and a

minimum of 30 V. Upon opening the main battery enclosure after the aircraft was towed into a hangar, traces of leaked electrolyte were observed inside the enclosure. After the events at Boston and Takamatsu in January 2013, the design of the battery and the battery charger unit was modified and a new enclosure was installed by Boeing. This event at Narita was the first in which smoke was observed from any in-service battery after these improvements had been incorporated. The voltage of the main battery was 27 V, which, while being lower than the nominal voltage of 31 V, meant that the main battery would be able to provide the required voltage for continued flight. Boeing decided to follow three principles in redesigning the electrical system. Three layers of improvement were incorporated: a first layer to prevent the cell from overheating; a second layer to prevent cell to cell propagation in case of cell overheat; and a third layer to prevent fire in case of cell to cell propagation. The 250-seat jetliner, which costs about $212 million at list price, had initial problems. It also had issues also its brakes, fuel lines, hydraulics, and other systems, but misfortune predominantly struck the battery systems, exclusively for Japanese airlines and in January!

In discussing electrical abuse, we should consider the case described in [48], which reported a fire in a PHEV which occurred while driving (fortunately, the driver was able to escape). Many curious aspects are described in the report of the accident. For example, that the melting temperature of a nickel sheet (1560 °C) was reached. This aspect allows us understand the risk of a thermal runaway. In this case, fortunately, the fusion of the collector sheet electrically separated that part from the rest of the cells in parallel. It must also be said that the battery pack used on the vehicle in question was an aftermarket, and not that provided by the PHEV manufacturer, and that the phenomenon was triggered by an electric arc generated by an incorrectly fastened bolt, i.e., a washer had been placed the wrong way around.

Previous abuse can cause malfunctions, with thermal consequences. In such cases, the temperature increases until the first cell, and subsequently, the neighboring ones, are destroyed. The authors of [40] describe the phenomenon in three steps: level I, the cell with an internal short circuit shows self-extinguishing features, i.e., there is slow self-discharge but no apparent heat generation; in level II, the characteristics of the internal short circuit become more clear, with a faster drop in voltage and a faster increase in temperature; finally, level III represents thermal runaway, with unstoppable heat generation due to the collapse of separator (SEI). Such a phenomenon was observed in the aforementioned aircraft accidents. Figure 11 summarizes the three levels described in [41].

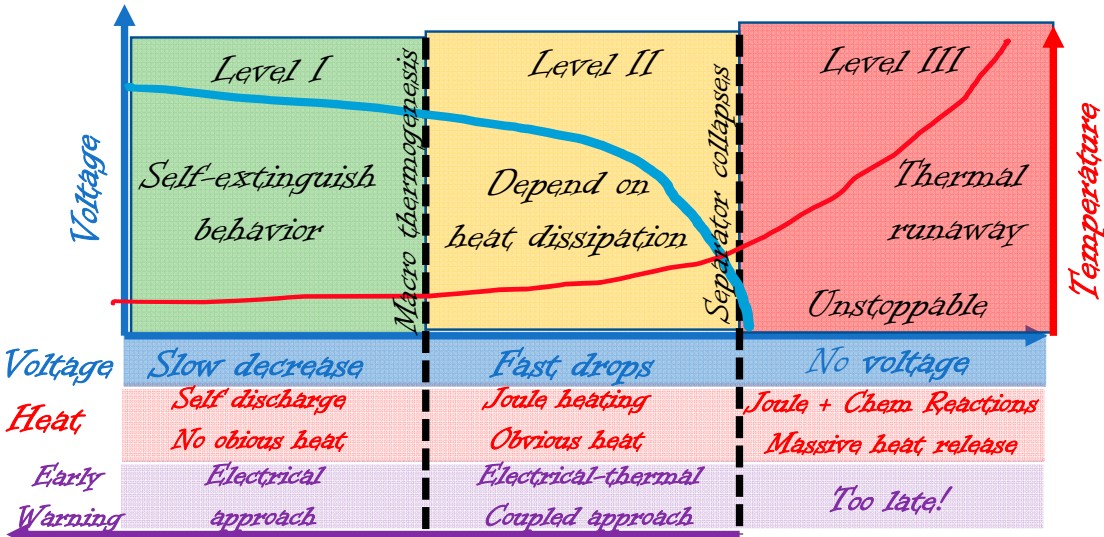

**Figure 11.** Different levels of malfunctions: between levels I and II, macrogenesis of heat occurs; between levels II and III, there is a separator collapse [41].

Auspiciously, the occurrence of spontaneous internal short circuits takes a long time to evolve from Level I to Level III, so the battery management system can interrupt the discharge before Level III is reached, and the car can be evacuated, which requires 30 s, after an accident.

In order to avoid the occurrence of thermal runaway, different safety strategies have been adopted, e.g., modification of the cathode materials, modification of the anode materials, more stable electrolyte systems, or the use of advanced separators. Some modern safety devices are [41]: Cell Vents or Tear-Away Tabs, allowing the safe release of gas if excessive pressure arises inside cells; a Shutdown Separator between the anode and cathode, preventing ionic conduction if the cell internal temperature exceeds a certain limit; Current Interrupt Device (CID), which protects against over-current by breaking the internal electrical connection when the internal pressure reaches a certain value; Positive Temperature Coefficient of expansion (PTC) disks, placed in the cell header to limit high currents; Current Limiting Fuses, used in place of PTC devices when a sustained discharge is not applied; Diodes, preventing a low SoC cell from becoming reverse polarized by a series higher SoC cells during a massive discharge (i.e., a bypass diode); Battery Management System (BMS), controlling the electrical distribution with a battery pack and protecting against over- or under-voltage conditions, as well as excessive current or temperature. It is estimated that 48% of the cost of an EV is dedicated to the battery, but it must be noted that it is not the cost of the cells themselves, but of the battery protection system that makes up a large part of this. Economy vehicles may not apply the three safety levels described above.

Based upon this information, users may be inclined to consider EVs to be unsafe. However, the self-induced failure of lithium ion batteries occurs very rarely. The authors of [41] reported that the failure rate is approximately 10,000 times less than that of traditional vehicles (7.6 fire accidents per 10,000 vehicles). This point was also noted in [40], since ICEVs, which dissipate a lot of power in the form of heat, are more likely to suffer from short circuits due to heat damage to insulators. The authors of [42] stated that if safety devices work well, the failure rates of lithium ion rechargeable battery cells would be less than 1 in 10 million, or even 1 in 40 million cells. The probability of an EV spontaneously catching fire without having been in an accident and with a modern safety system, and subsequently exploding, is close to that of dying from a local meteorite impact or having a car accident with a great white shark, as suggested in [49]; see Table 1.

**Table 1.** Fear of explosion: table of odds of dying from selected causes in a human lifetime. The probability of an electric vehicle explosion is the same as being hit by a meteorite, or bitten by a shark.

| Odds of Dying from Selected Causes in A Human Lifetime | | | |
|---|---|---|---|
| Cause | Odds | Cause | Odds |
| Motor Vehicle Accident | 1 in 90 | Lightning strike | 1 in 43,000 |
| Suicide | 1 in 120 | Asteroid Impact Global | 1 in 75,000 |
| Homicide | 1 in 185 | Terrorism (non Middle East) | 1 in 80,000 |
| Falls | 1 in 250 | Tsunami | 1 in 100,000 |
| Terrorism (Middle East) | 1 in 1000 | Insect bite or sting | 1 in 100,000 |
| Fire or smoke | 1 in 1100 | Earthquake | 1 in 135,000 |
| Electrocution | 1 in 5000 | Asteroid Impact Regional | 1 in 1,600,000 |
| Drowning | 1 in 9000 | Food Poisoning by Botulism | 1 in 3,000,000 |
| Flood | 1 in 27,000 | Shark Attack | 1 in 8,000,000 |
| Airplane Crash | 1 in 30,000 | EV explosion | 1 in 10,000,000 |

## 6. Viking Men Paradox, or Gender Attitudes

In the definition of the profile of first adopters of EVs, some psychological and socio-economic aspects were highlighted. Although a dualism was found between an eco-friendly attitude and purchasing behavior, the transition from ICE to EV was shown to often be conditioned by general well-being (income, household size, ownership of more than one car) as opposed to education, gender, or age, although some common recurrent perceptions

were found. In the U.S., currently, the number of men with EVs is twice that of women [50]. This is a good reason for deepening the investigation into gender attitude.

The authors of [37], in discussing the main attitudes toward EVs in the early years of the twentieth century, stated that "women preferred to push an electric button than turn hand cranks to start", identifying EVs as girlish, in contrast to ICEVs, which required physical prowess (more a matter of cleanliness and hygiene, or quick reflexes, if the crank escaped, we suppose). Articles in a popular magazine from around the same time, as cited in [51], asserted the same thesis: Phil A. Riley suggested that "EVs were perfectly suited to the needs of women, travelling shorter distances, near to home, needing an ever ready runabout for daily use, leaving extended travels and fast driving to the men in gas powered cars." C. H. Claudy, also stated that EVs suited to women since "the car which has a circumscribed radius", "a machine which she can run herself, with no loss of dignity", "in no way can a child get so much air in so little time as by the use of the automobile, to call the electric the modern baby carriage" and also for a matter of hygiene compared to the grooming horses. Luckily, such ways of thinking have been overcome, even with demonstrations by adventurers like Emily Post and Alice Huyler Ramsey, responsible for pioneering ventures.

But what are the actual gender perceptions of electric vehicles? Different articles have indicated that environmental issues and climate change are more important to females than males [9,10,16,50,52].

In [53], the author reported that women are more sensitive to environmental issues and more willing to reduce their car use than men for sustainability reasons. Men are more sensitive to power and performance, but are also inclined to cycle in "unsafe" places to signify confidence and bravery [52].

The authors of [54] suggested that "males have a higher preference to purchase EVs than females", and that "males tend to be much more interested in the latest technological items than females". This vision depends on many aspects, since, as shown in [9,10], women are more interested in EVs than men.

A common stereotype is that men never ask for directions, i.e., "men find it hard to ask for help because it is a submissive gesture" [51]. In [55], it was reported that the German engineers who designed navigators for BMW insisted that the computer have a male voice. The engineers reasoned that "men don't want women giving them directions".

Savacool performed a very interesting analysis regarding gender identity in an international survey [52]. The answers to the questions were curious. In order to define the desires of younger drivers, one respondent replied "Most boys want to drive big, fancy cars, or trucks, for going into the country" (see the desire to reach the frontier, in the Range Anxiety section). "Nobody wants small car, especially those seeking to be macho Viking men". A female participant commented that, thinking about her parents, her dad wanted a "huge-back station wagon", while her mom had a little car to go to work and stuff. Her father had no need to go on long range journeya with the family and a pack of dogs, he had no farm to manage, but for him, men need a big car. Savacool [16] returned to the Viking man myth; here, we again report the opinions collected in his works. "in a traditional bourgeois household, the man is driving the big car [and] the woman maybe works half time and has the small car. Now it is shifting around. The man has the small car to get to work every day, the battery car, the woman has to drive the kids to football practice and to school and so on and needs a bigger car for that".

However, if we consider the spread of EVs, it is necessary to note the following. In [50], it was stated that in Maryland, there is a big gender gap in EV ownership, but it is possible that most households registered their EVs in the name of a male family member. A smart vision was provided by a female energy expert in Finland [51], stating that although her husband had bought the dishwasher and washing machine, she was the one actually using them.

In order to better address the issue, and to investigate these issues more deeply, it is advisable to consider a Norwegian study, a country in which there was a transition from

early adopters to the early majority [56]. In this article, a survey was undertaken which was based on interviews with electric vehicle owners, aiming to define gender attitudes. It was found that different factors attracted EV users in Norway besides environmentalism and economy. The report stated that EVs appear as a symbolic hybrid, with both feminine and masculine connotations. It was found that there was an inclination between both men and women to emphasize that men drive more often and for longer distances.

Table 3 reports some of the gender attitudes found in the survey. Attitudes are divided into symbolic, practical or cognitive.

**Table 2.** Report of different opinions between females and males about vehicles and, specifically, EVs. Couples have been highlighted in order to show the different ways of thinking. Opinions are separated into symbolic (sand), practical (green) and cognitive (blue) issues [51,52,55,56].

| Gender | Symbolic Practical Cognitive |
|---|---|
| Male | Fascination for German cars |
| Female | No such specific interest in cars |
| Male | Did the math comparing gasoline, hybrids and Evs |
| Female | Consider costs of insurance, annual road fees, gas and tolls |
| Female | Studied the Tesla in car magazines and took it for a test drive |
| Female | Not involved |
| Female | Energy efficient and economic driving |
| Female | Range was prioritized over comfort, while winter coats can be employed to keep bodies warm and batteries long-lasting |
| Male | Technical 'motor-related' interest |
| Female | Green lifestyle |
| Male | EVs are easier to control, enhancing manoeuvrability |
| Male | How fast the car goes from 0 to something in 4 s |
| Female | Transport me comfortably from A to B |
| Female | EVs have different limits, and are faster |
| Male | I'm noticing that she uses way more electricity than me, but that's most likely because I've driven it more |
| Female | He is a bit more used to it |
| Female | Evs primarily benefit the environment |
| common | The second car has traditionally been "the wife's car" in Norway |
| common | Dad drives the big, nice, new diesel car, while mom drives the little, old one |
| common | Smaller cars don't appeal to men |
| Female | Men are concerned with the more mechanical aspects of cars |
| Female | Women focus on comfort and easy manoeuvring |
| Male | EV evolution is like going from a Nintendo Entertainment System to a PlayStation 3 |
| Male | New type of driver: electric motorist |
| Male | Tesla appealed more to men because it symbolized power, speed and status |
| Female | Environmental profile of the Tesla suited to women |
| Female | People here really like their big Jeeps and the idea of the freedom that they represent, also to go to the bakery |
| Male | The most common EV in the Nordic Region is a Tesla . . . It is a beautiful car, cool to have |
| Female | satisfaction of "Viking" identity to drive big, fancy cars, or trucks, for going into the country. There are all these small cars but nobody buys them, especially those seeking to be macho, i.e., "Viking men" |
| Female | Most mechanics are men; it's a very male-dominated branch. Men are the decision-makers around the house, and most car salespersons are men |
| Female | When I think about my parents, it was always my dad who wanted a huge-ass station-wagon and my mom who got a little car because she also needed to go to work and stuff |
| Male | Men find it more difficult to switch to cleaner or small cars, and women can switch more easily |
| Male | My girlfriend, for example, likes a small car that she can park easily |
| Male | Who likes big cars that make a lot of noise, go really fast and are super nice and comfy? Men. Who is environmentally-friendly and likes small cars? Then you're girly and more feminine. |
| Male | EVs are effeminate and environmental |
| Male | A woman's car is red, safe, and kind of small, and it drives around the city. It will be child-friendly and stuff like that, and usually with room for a dog |
| Male | Men want something which goes fast, with flames or naked women, and driving through a mountain area |

**Table 3.** *Cont.*

| Gender | Symbolic Practical Cognitive |
|---|---|
| Male | EV owners see characteristics such as economic issues, range and public recharging as less important than participants who do not own an EV |
| Female | stronger preferences than males regarding specific attributes like range, battery life, public charging and charging time |
| Male | If you want to go with a blonde, you want a car with acceleration. And electric cars, they have very good acceleration |
| Male | If you use a Buddy, you immediately look eighty years old |
| Male | In a traditional bourgeois household, let's say the man is driving the big car, the woman maybe works half time and has a small car. Now it's shifting around. The man has the small car to get to work every day, the battery car, the woman is driving the kids to practice and to school and football and so on and needs a bigger car for that |
| Female | the one at home who's actually dealing with the daily energy system is probably the woman. My husband bought the dishwasher and the washing machine, but I'm the one actually using it. |
| Female | My father wanted a huge-ass station wagon, even if he did not need to go on long range journeys with family and a pack of dogs |
| Male | The sad part about electrical cars is that they don't make any noise, and the noise is the sexiest part of the car |
| Female | (about Tesla) It's a real housewives' car. You can put all the groceries in the back, and your handbag between the seats in front. If you haven't bought the stupid centre console, you can do that, at least. That's what women wanted: a place to put their bag. |
| Male | Men find it hard to ask for help because it is a submissive gesture |
| Male | Men don't want women giving them directions |
| Male | Most people wouldn't have the patience to drive EVs since of all the brain power was used to plan the trips |

## 7. Autonomous Silver Vehicles

The world has never seen so many people over seventy, called the baby boomers. This is called the "Graying of Society", and is an actual megatrend that human society is facing or is about to face [57]. As people grow older, they have a tendency to use more public transport as an alternative to driving their vehicles. As reported in [58], the United Nations estimates that the older populace of the world will increase from 962 million in 2017 to 2.1 billion by 2050, reaching 3.1 billion in 2100. Such a population growth is never accompanied by an equivalent development of transport systems, both for economic reasons and logistic ones, since cities are exploding but also imploding. Radial expansion is accompanied by development in height. In the early morning, there is a convergence from the periphery towards the city center, while in the afternoon rush hours, there is a departure from the city towards the periphery, creating a so-called donut effect. In such a situation, it is possible that traffic jams, extending for kilometers and taking hours to pass, may arise.

Not conditioned by work problems, the elderly are generally considered vulnerable road users [59]. The barriers preventing the elderly from driving are distinct; in [58], they are grouped follows: health, environmental, economic and social factors. Health factors include physical, psychological and cognitive issues. Physical issue includes limits in the ability to walk, cycle, drive, see and use also public transport services. The psychology of ageing hinders the ability to drive, since there is a fear of being stuck in a traffic jam or involved in crash. Cognitive limits regard the difficulty of using technology or interpreting maps. Environmental factors are related to new road design, with large or high-speed intersections being intimidating to the elderly. Economic factors limit the use of taxis. Social factors are related to the difficulty of leaving a familiar place, for example, the refusal to abandon a little, old grocery shop in favor of a shopping mall.

One possible solution to the aforementioned barriers is silver self-driving vehicles, as noted in a survey presenting the opinions of senior citizens regarding different scenarios in the province of Utrecht, in The Netherlands [60]. Four scenarios were studied: (1) automated public transport with fixed schedules and routes, which employs high occupancy vehicles (50 or more person) with fixed stops (similar to a bus system, but with no driver); (2) automated on demand public transport, with low occupancy (6–14 persons); (3)

fleet-based automated shared vehicles, which offer carsharing for the family or ride sharing with strangers travelling similar itineraries; (4) privately owned automated vehicles. The survey highlighted various results: (a) most of the participants showed a strong preference for on-demand scenarios (2, 3 and 4); (b) the shared solutions made it possible to socialize when not traveling with family and friends; (c) the participants expressed the concern that high cost could limit frequent use to visit family and friends; (d) a complete lack of confidence in the automated driving system was highlighted.

A similar survey was presented in [58] concerning futuristic scenarios whereby costly autonomous transportation establishes a dualism between autonomous vehicles (AVs) and roads (including traffic management). Scenario 1: Private AV (PAV), i.e., a costly vehicle owned by a family, the sharing of which depends on willingness of the family; as main roads will host unused vehicles, limiting transport, there will be a possible decline in public transport. As such, old people will prefer to own their own AV. This scenario is a transposition of the hegemonic present into a more technological future. Scenario 2: Unconstrained shared fleet, private ride-sourcing companies offering taxi services; the roads become a commodity for companies, some roads are accessible to a given company, others are not, and disputes arise between local authorities that see private companies as competitors for the use and regulation of their roads. A similar scenario presents economic risks for the use of the service by the elderly. Scenario 3: new demand management, in which PAV and shared AV (SAV) of corporatized companies coexist; governments regulate the access of roads and PAV should become SAV to reduce congestion; the government protects the elderly by regulating access to the service. Scenario 4: Public Mobility as a Service (MaaS), a radical public management approach in which AVs exist only within a government-managed public service, i.e., a dynamic ride sharing system, which offers tailored services for older people. Different aspects arise from these hypothetical scenarios. Except for the first scenario, there will be a revolution in transport systems with increasingly reduced freedom to move where you want. The use of some roads instead of others could also affect the movement from A to B, and change the habits and purchasing attitudes of users.

The roadside structures, communication systems, safety and security issues and business models which are required in order to implement such solutions are fully explained in [57]. While it seems that public transport such as trains may decline, given the services tailored to the elderly, this cannot be said for their technology. The movement of autonomous vehicles is not possible without the contribution of the most advanced technologies, especially telecommunications. To solve the problem of the limited capacity of existing roads, the absence of side parking spaces is assumed, but this is not enough, it is necessary to increase the travel speed of the roads themselves. What may seem like an ambitious goal can be achieved by inheriting the concept of wagons and moving blocks from trains. The journey of a vehicle from point A to point B can be divided into subroutes, for which it is possible to find affinities with other vehicles. Assemblies of vehicles with similar destinations are constituted like wagons of a train, comprising about fifteen vehicles, which march compactly like a platoon. Different platoons are addressed in the use of roads with the technology of moving blocks of trains, for which there is always the prediction of the behavior of the previous and subsequent blocks, to reduce the possibility of impact to zero. Different levels of communication are therefore required: the first is peer to peer, i.e., between vehicles to ensure the formation of the platoon. The next level is given by the dialogue between platoons and roads, so that each platoon does not run any red traffic lights; therefore, communication is also assumed with the roadside. Finally, a hierarchically superior system establishes the means by which to aggregate the platoons according to the needs of each vehicle.

Although autonomous driving systems are widespread, a distinction must be made between levels of autonomy from zero (no interlocking) to full autonomy, i.e., level 5 (no steering wheel!). Level one is represented by the possibility of correcting maneuvers in dangerous situations (wheel spin, skidding, crossing the middle of the road); level 2 sees

the vehicle itself performing a maneuver (parking); in level 3, the vehicle is given the opportunity to drive long distances by overtaking and crossing intersections, but the driver must always be ready to take command of the situation.

In level 4, the vehicle can proceed without the driver being in the driving position, and in level 5, there will be no driving position at all.

A consequence of giving greater responsibility from the user to an autonomous system is increased cost.

For the realization of such a system, the technology is ready, but there is a lack of infrastructure, the financing of which is unfeasible for individual municipalities. Autonomous driving must interact in a vehicle-platoon-roads system; therefore, a pervasive telecommunications system is necessary, which no car manufacturer can afford. Perhaps we will witness vehicles with their own system, as in the case of Apple, which creates both software and hardware, but in other cases, we will see vehicles with third-party software, as in the case of phone systems that use the Android platform. These scenarios also require new business models, since the change will be so radical as to revolutionize insurance systems, but also regulatory systems, as vehicles with obsolete software or without autonomous driving will no longer be able to circulate. But they also promise to reduce accidents.

It was therefore found that regardless of future scenarios, there is a population target which is inclined toward accepting the automation of vehicles. Contrary to what one may think, i.e., that the elderly are less likely to change, the propensity to keep their habits and comforts will push them to accept self-driving vehicles, especially in order to be liberated from responsibility. The autonomous vehicles of the future could be called silver vehicles, due to the strong propensity to use them by the elderly.

Baby boomers, in order to avoid the traumatizing experience of driving in traffic jams, will demand safer and more autonomous transportation. But what will be the attitude of the baby boomers' grandchildren?

In [57], it was reported that for the younger generation, owning a new tablet or phone is more desirable than driving a Porsche 911 at 20 km/h in city streets, so the baby boomers' grandchildren will embrace the shared modern EV more willingly than their grandparents.

## 8. Marriage or Cohabitation?

Several scientific studies have proposed that to overcome doubts regarding the performance of EVs, experience on the road is useful [9,10,61–67]. Electric vehicles are often seen as evolved golf-karts, but experience can demonstrate how much fun it is to drive one, and that it doesn't perform less well than an ICEV [63]. Common perceptions regarding EVs are: the limited range, difficulties associated with charging, difficulties of refueling at home, "how they see me" (perception of others), driving fun, and convenience.

Regarding the range issue, an interesting study is presented in [64], in which drivers were forced into critical range situations in order to learn from them. Range anxiety may be divided into range competence, range appraisal—including primal appraisal, which is a challenge, or even a threat—and range stress, i.e., when initial challenge feelings are surpassed by threat feelings. Two trials were performed: in the first, there was a SoC to end the trip, and in the second there was not. Both tests reported good adaption effects. The study presented in [65] reported unsatisfactory experiences regarding range, and it was noted that careful planning was required for journeys of distances greater than 30 km. A very different experience was reported in [66]. After one year of leasing an EV, drivers stated that they had more positive feelings and higher buying intention regarding such vehicles. Reporting their impressions, different people appreciated the regenerative brake "I stopped using the brake, I think I might use the brake once a month", "It is just been kind of fun to play with it and drive it, and it has a lot of zip and a lot of energy" "I love that I rarely use the brakes. You stop where you want to stop without using your brakes", "You can basically drive with one foot ". In another case, a subject observed that "Most people wouldn't have the patience to drive EVs since of all the brain power was used to plan the trips", like doing homework!

The initial difficulties in using refueling stations were overcome, and the process was found to be much cleaner than using a gasoline gun. Ideas on how to improve charging stations are presented in [29]; in that study, a comparison is made on where to place charging stations according to BEV users and interested nonusers. Both categories agreed that fast charging stations must be present at gas stations. Again, do BEV users not believe in higher education, as found in the previous paragraphs? At least there was confirmation that more fast stations are required near educational institutes (71% of female intervieweese said that this would be useful, compared to 55% of male interviewees). In a gender survey, the difference was reported among users regarding "Willingness to vacate the parking lot"; 47% and 36% of male and female users, respectively, expressed agreement while 36%, while for "Charging stations should be well-lit", 63% and 83% of males and females, respectively, expressed agreement.

The topic of refueling at home was addressed in [65] and enthusiastic opinions were found, e.g., the most significant EV benefits were the convenience of home recharging and reductions in average travelling cost.

The subject of reputation was treated in [9,10,62]. In order to better analyze the reasons for owning a BEV, it is necessary to consider both overcoming the need to move from point A to point B, called the functional aspect, as well as the symbolic aspect. Owners are proud to differentiate themselves from others, to show their lifestyle and show environmentally-friendly feelings, but also to be innovators of new technologies [10].

Finally, the most surprising aspect of the trial experience for everyone was the driving itself. The stereotype of golf karting was dispensed with and doubts about acceleration were dispelled; indeed, acceleration was one of the most appreciated aspects. Even regenerative braking was found to be pleasant [66].

We therefore conclude that in taking an important step like that of switching to a BEV, the initial insecurity can be dispelled with adequate testing, just like living together before marriage, Figure 12. The partner always met the expectations, although some complained about the need to do homework to evaluate routes or were afflicted by range anxiety, and particular appreciation was shown regarding overcoming the noise barrier [65,66], with less than 3% declaring this aspect to be a barrier to buying such a vehicle [61]. Smart people avoid the absence of a mental commitment and noisy partners, and prefer intelligent and quiet partners.

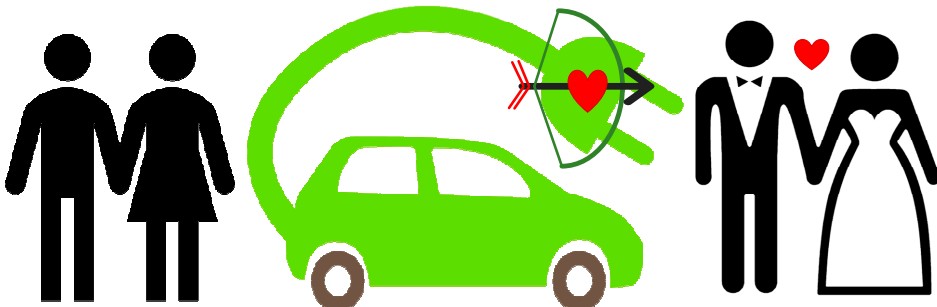

**Figure 12.** Marriage after cohabitation [65]; it was found that 59.2% of subjects would be willing to purchase an EV after the trial.

## 9. Conclusions

This review reported data present in the technical literature and discussed the most curious aspects that fall within the spheres of our daily life. The main attitudes influencing the transition from a conventional ICEV to BEV were presented; limiting points of view include the inertia to abandon a hegemonic present, to embrace a future vision. A first goal was to consider the reactions of users toward this new product. Some studies described the characteristics of early adopters; however, surveys were not able to define common characteristics, other than the medium-high position regarding income. This aspect is also due to the fact that, in the embryonic stage EVs, cost more than ICEs, and are not very

popular second hand. Very curious was a description of users—in some cases, people over 50, and in others, young people—who were worried about how others would perceive their environmentally-friendly behavior. Notably, in Norway, early adopters represented a majority. Norway's success was compared to the failure, regarding a lack of leadership in the diffusion of EVs, of Sweden, a nation accustomed to innovation. This indicated which policies should be avoided, such as the use of underground charging stations. Speaking of charging stations, the problem of the dualism of the egg and the chicken was addressed. Situations were shown in which the first early adopters suffered from disadvantageous circumstances such as low numbers of charging stations; however, these studies also presented data regarding the use of charging stations, showing differences between countries based on the vehicle recharging coefficient index (VRI) and the presence of stations along highways. This is necessary to minimize the fear of being stuck in a blizzard with a flat battery. Such fear falls into the broader anxiety range. This anxiety was discussed and an attempt was made to evaluate it numerically, demonstrating that although similar situations can occur a few times per year, it is still one of the main obstacles to the use of EVs, precisely because of the theory of the reaction to innovation, which does not allow one to abandon the certainties of the present. Among the false conditions for the adoption of EVs, there is the fear that they will give rise to unstoppable fires, and that all EVs are therefore dangerous. The most famous cases of malfunctions in the technical literature were cited, showing the paradoxical situations in which such accidents occurred, from the replacement of the original battery pack with one that was not properly installed and for which the melting temperature of a nickel sheet (1560 °C) were surpassed, to the very destructive tests in which not even the driver would survive, and the unfortunate sequence of malfunctions in planes built by Japanese companies. Protocols were discussed for dealing with cell failures which prevent the phenomenon of thermal runaway, and when combined with thermal technology, make the likelihood of an accident akin that of being hit by a meteorite.

The main attitudes towards EVs were discussed, including the ways in which such vehicles are perceived in northern European countries, and the opposition between Buddy (city car) and Tesla. Small EVs were initially considered to be girly or for octogenarians by users falling within the category of "modern Viking men". The advent of Tesla connected the male and female worlds: for enthusiastic males, the Tesla was a technological advancement akin to that from a Nintendo Entertainment System (1983) to a PlayStation 3 (2006), while for enthusiastic women, it was a refined place to put the handbag. The advent of autonomous EVs was also discussed. In this respect, a comparison between generations was presented; it was found that older users, in order to keep their attitudes, will embrace the new self-driving EVs, named silver vehicles to remember the possibility of helping them with the weaknesses of age, but also their grandchildren thinking that it is more important to have a latest generation mobile phone than to drive a Porsche 911 in the city, will appreciate the revolution of shared and autonomous mobility.

Finally, EV user testimonials were reported, which were always positive, although complaints included the difficulty of doing homework (calculating charging stops during journeys) and the absence of noise, which is considered by some to be the sexiest part of a vehicle. The driving experience was one of the most surprising aspects. The stereotype of golf karting was dispensed with, and a new category of drivers can be identified: electric motorists!

**Funding:** This research received no external funding.

**Institutional Review Board Statement:** Not applicable

**Informed Consent Statement:** Not applicable

**Data Availability Statement:** Data sharing is not applicable to this article.

**Conflicts of Interest:** The authors declare no conflict of interest.

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
