# Peer review of "Electric Vehicles and Psychology"

_sustainability, doi:10.3390/su13020719_

Round 1

Reviewer 1 Report

Dear author,

first of all, I would liek to thank you for an intersting work dealed with an actual topic.

In this review the chosen psychological attitudes influencing the transition from conventional ICEV to BEV were presented.The review called "Electric vehicles and psychology" is logical and clearly devided in to 9 chapter hwere the first one is Introduction and the last one Conclusion. the middle lockated chapter are sovling issues connected with electric vehicles implementation like f.e. "early adopters", "egg or chicken paradox", range anxiety and others.

The appropriate and adequate references to related and work were used and the number of used referneces for such review is appropriate.

Contradictions and irony characterized this paper (like the author wrote also in the abstract). There is also scientific soundess but the irony, contradictions, sarcasm and unprofessional terminology is used in high frequency. This fact significantly decreases the scientific value of this review.

Reading of this review was really an interesting experience and I enjoyed it. I subscribe to this author´s point of view - irony and analyzing burning topics, swimming against the current, expressing oneself not quite correctly... - these points make the review very interresting and engaging. But opposite this facts - these points, matter of writing, make also this review inappropriate for publishing in a scientific journal.

I suggest to the author to erase inappropriate expressions and pictures, rebuild sentences (journalistic writing style) in to more scientific soudness.

These steps could help to make the review more scientiffic and leave here the ironic approach to this problematic.

I really would like to read this review one more time after publishing it in this actual form. But it should be published in some casual popularization-technical publication, not in a scientific journal.

I suggest to rewrite the review and try to resubmit it - also maybe better to an scientific journal oriented for transportation sciences, behavioral sciencec or psychology.

Some suggestions:

  1. The first section of the Introductiona should be rewrite.
  2. Chapters 2 and 4 are written in the most appropriate approach.
  3. Pictures 3, 4, 7, 9, 10, 11, 12 are not suitable for a scientific work even it is ironic written. Some of them have at least connection to the solved topic, but pictures like 3, 7, 9 give a hint of infantility.
  4. Please, avoid the matter of writting used f.e. in the first section of the chapter 4 (rows 261-266).
  5. Please, avoid the expressions like f.e. "huge-ass station wagon" used in row 540.
  6. Please, expand and deepen the Conclusions, it is too short and inadequately.
  7. Please, aviod using the journalistic and layman´s style of writing in whole work.

Author Response

First of all, I would like to thank you for an intersting work dealed with an actual topic.

Author kindly thanks the reviewer for the time devoted to the study of the paper.

In this review the chosen psychological attitudes influencing the transition from conventional ICEV to BEV were presented.The review called "Electric vehicles and psychology" is logical and clearly devided in to 9 chapter where the first one is Introduction and the last one Conclusion. the middle lockated chapter are solving issues connected with electric vehicles implementation like f.e. "early adopters", "egg or chicken paradox", range anxiety and others.

The appropriate and adequate references to related and work were used and the number of used references for such review is appropriate.

Contradictions and irony characterized this paper (like the author wrote also in the abstract). There is also scientific soundess but the irony, contradictions, sarcasm and unprofessional terminology is used in high frequency. This fact significantly decreases the scientific value of this review.

Thank you for the point of view, the paper has been revised in order to reduce irony and add more scientific point of view. The reviewer can find novel analysis and figures that add the lacking scientific aspects in little parts of the paper.

Reading of this review was really an interesting experience and I enjoyed it. I subscribe to this author´s point of view - irony and analyzing burning topics, swimming against the current, expressing oneself not quite correctly...- these points make the review very interresting and engaging. But opposite this facts - these points, matter of writing, make also this review inappropriate for publishing in a scientific journal.

Thank you for the point of view, the paper has been revised in order to gain the publication. I think that in a special issue, between articles facing inverters, technical economical aspects, life cycle cost analysis, a review of EVs and psychology is an added value. This paper cannot be published in a IEEE industrial electronics transaction, but in a special issue of Sustainability gains a right sounds. I hope the Reviewer changes his mind.

I suggest to the author to erase inappropriate expressions and pictures, rebuild sentences (journalistic writing style) in to more scientific soudness.

The point of view has been followed. Pictures were deleted or used inside other pictures, sentences were remodeled and sometimes were reported with the reference to scientific journal in which where found.

These steps could help to make the review more scientiffic and leave here the ironic approach to this problematic.  I really would like to read this review one more time after publishing it in this actual form. But it should be published in some casual popularization-technical publication, not in a scientific journal.

The author thanks the Reviewer for his point of view, but firmly believes that a special issue of an excellent magazine, in which the author has already published a profiling article of adopters of fuel cell technology vehicles, could also host articles in which he makes fun of the failures in the perception that a community has of electric vehicles when it does not rely on scientific aspects. However the article has been profoundly changed from its original version, bringing back many more scientific graphs.

I suggest to rewrite the review and try to resubmit it - also maybe better to an scientific journal oriented for transportation sciences, behavioral sciencec or psychology.

Some suggestions:

  1. The first section of the Introductiona should be rewrite.

The suggestion has been followed. The introduction was rewritten and the paradoxes are apprached with different questions.

  1. Chapters 2 and 4 are written in the most appropriate approach.
  2. Pictures 3, 4, 7, 9, 10, 11, 12 are not suitable for a scientific work even it is ironic written. Some of them have at least connection to the solved topic, but pictures like 3, 7, 9 give a hint of infantility.
  3. Please, avoid the matter of writting used f.e. in the first section of the chapter 4 (rows 261-266).
  4. Please, avoid the expressions like f.e. "huge-ass station wagon" used in row 540.
    Please, expand and deepen the Conclusions, it is too short and inadequately.
  5. Please, aviod using the journalistic and layman´s style of writing in whole work.

Suggestions were followed. Paper was deeply revised, figures were removed and new more scientific were used. Some way of writing were removed and represent on a table. Conclusions were rewritten. Reviewer can find all in red color in the revised version.

The author sincerely thanks the Reviewer and believes that this choice was suitable, since he understood the spirit of the article, which is not easy

Reviewer 2 Report

My main problem with this submission is the author is ignoring the most important issue which is the cost of ownership and running cost of the vehicle.

Although the range is extremely important it is dependent to the cost of vehicle.

Other issued referred to in the paper are already known and discussed. I can not see what is the contribution in this paper.

The paper needs more analytical content.

Author Response

main problem with this submission is the author is ignoring the most important issue which is the cost of ownership and running cost of the vehicle.

Author kindly thanks the Reviewer for the time devoted to the study of the paper. The total cost of ownership is not a psychological issue. Author did not face it since it is too analytical, but different  websites of car manufactors (i.e. https://www.jeep-official.it/4xe-ibrido) evaluate the benefits of the ownership by running a quantity of kilometers by year. In the first part of the review, (finding early adopters) different features were considered, and always adopters have a high income. So author thinks that this issue is important but it should be faced in a section,  not an own one. However in the review of the article it was included end more addressed. The Reviewer can find them in red color or with new and more suitable figures.

Although the range is extremely important it is dependent to the cost of vehicle.

Author thinks that this is a good point of view. In the last years different are the models with a very long range batteries, and very high prices. Authors considers that even if an EV can cover a distance that drivers do in 360 days of the year, for the last 5 days of the year the EV'range will be seen as limitated . So an EV should never sotisfy the driver.

Other issued referred to in the paper are already known and discussed. I can not see what is the contribution in this paper.

Author thanks the reviewer for his point of view, but he struggled to make many concepts that are considered widespread present in a synthetic review, also because he chose to include sentences taken from high-level scientific journals. The author finds it useful that these concepts are present in a single scientific article.

The paper needs more analytical content.

The paper was revised in such point of view. More detailed analysis were added and new figures found place to enforce the analysis. 

Author thanks the Reviewer for his time devoted to the study of this paper.

Round 2

Reviewer 1 Report

Dear author,

thank you for the revised manuscript. I can see there an improving mainly in the scientific soudness of the work.

I think, that this manuscript needs minor revision before publishing.

Fig. 6 should be removed, it is not suitable for a scientific work.

Fig. 12 should be remaden, it gives still hint of infantility (the right side of the figure).

Fig. 13 is reduntant is present form, I suggest to remake it into a shorter form with a higher (fast, flash) informative value after first look. It is not suitable to introduce long and detailed outputs of a referenced research results.

I still suggest to avoid expressions in the manuscript like f.e.: "My cousin told me...." and similar others.

Author Response

Dear author,

 thank you for the revised manuscript. I can see there an improving mainly in the scientific soudness of the work.

I think, that this manuscript needs minor revision before publishing.

The author greatly appreciates the contributions of the Reviewer that have given the possibility to change his article for the better.

Fig. 6 should be removed, it is not suitable for a scientific work.

Figure 6 has been modified and the source has been provided. Two figures were taken from a technical report of Columbia university on page 8,

https://energypolicy.columbia.edu/sites/default/files/file-uploads/EV_ChargingChina-CGEP_Report_Final.pdf

28 Anders Hove, David Sandalow, Electric Vehicle Charging in China and the United States, Center on Global Energy Policy at Columbia University SIPA 1255, Amsterdam Avenue, New York, NY 10027 (212) 853-2475, February 2019, www.energypolicy.columbia.edu

In this review the more curious aspects are taken from scientific publications!!

Fig. 12 should be remaden, it gives still hint of infantility (the right side of the figure).

Figure 12 has been changed as suggested

Fig. 13 is reduntant is present form, I suggest to remake it into a shorter form with a higher (fast, flash) informative value after first look. It is not suitable to introduce long and detailed outputs of a referenced research results.

I am sorry but I cannot condense the opinions reported in a different container. They are too greedy not to be presented to the scientific community, and then the paper will be online, and it will not damage the fluency of such a figure

Reviewer 2 Report

It is a better version than the original submission.

Author Response

It is a better version than the original submission.

Thank you so much for the help in improving the paper and time devoted in read a long review.